# Revealing and Harnessing Tumour-Associated Microglia/Macrophage Heterogeneity in Glioblastoma

**DOI:** 10.3390/ijms21030689

**Published:** 2020-01-21

**Authors:** Yolanda Pires-Afonso, Simone P. Niclou, Alessandro Michelucci

**Affiliations:** 1Neuro-Immunology Group, Department of Oncology, Luxembourg Institute of Health, L-1526 Luxembourg, Luxembourg; yolanda.piresafonso@lih.lu; 2Doctoral School of Science and Technology, University of Luxembourg, L-4365 Esch-sur-Alzette, Luxembourg; 3NORLUX Neuro-Oncology Laboratory, Department of Oncology, Luxembourg Institute of Health, L-1526 Luxembourg, Luxembourg; simone.niclou@lih.lu; 4Department of Biomedicine, University of Bergen, N-5007 Bergen, Norway; 5Luxembourg Centre for Systems Biomedicine, University of Luxembourg, L-4365 Esch-sur-Alzette, Luxembourg

**Keywords:** glioblastoma, tumour-associated microglia/macrophages, cellular heterogeneity, immunotherapy, precision medicine

## Abstract

Cancer heterogeneity and progression are subject to complex interactions between neoplastic cells and their microenvironment, including the immune system. Although glioblastomas (GBMs) are classified as ‘cold tumours’ with very little lymphocyte infiltration, they can contain up to 30–40% of tumour-associated macrophages, reported to contribute to a supportive microenvironment that facilitates tumour proliferation, survival and migration. In GBM, tumour-associated macrophages comprise either resident parenchymal microglia, perivascular macrophages or peripheral monocyte-derived cells. They are recruited by GBMs and in turn release growth factors and cytokines that affect the tumour. Notably, tumour-associated microglia/macrophages (TAMs) acquire different expression programs, which shape the tumour microenvironment and contribute to GBM molecular subtyping. Further, emerging evidence highlights that TAM programs may adapt to specific tumour features and landscapes. Here, we review key evidence describing TAM transcriptional and functional heterogeneity in GBM. We propose that unravelling the intricate complexity and diversity of the myeloid compartment as well as understanding how different TAM subsets may affect tumour progression will possibly pave the way to new immune therapeutic avenues for GBM patients.

## 1. Introduction

Gliomas represent approximately 80% of all malignant tumours of the central nervous system (CNS) [1]. Among them, glioblastoma (GBM) is the highest-grade glioma (grade IV) and the most common malignant brain tumour in adults. The standard care of treatment for GBM relies on maximal surgical resection followed by radiation therapy and concomitant chemotherapy with the alkylating agent temozolomide as established in 2005 [2]. However, recurrence is inevitable, and prognosis remains poor with a median survival of 15 months after diagnosis. Hence, the development of novel therapeutic options, including immunotherapies, are needed.

The immune landscape of brain tumours is intensely investigated, unveiling new insight in the interactions between neoplastic cells and the immune system [3]. GBM is a highly immunosuppressive cancer, where resident microglia and peripheral infiltrated macrophages play a key role in immune escape mechanisms [4]. Tumour-associated microglia/macrophages (TAMs), which can constitute up to 30–40% of the bulk tumour mass, outnumber by far infiltrating lymphocytes in these tumours [3]. This scarcity of lymphocytes in the tumour microenvironment contrasts with other tumour types, e.g., melanoma or lung cancer, therefore classifying GBM as immunologically ‘cold tumours’. Whether or not these tumours are intrinsically non-immunogenic or whether lymphocytes, including T cells, are actively excluded, remains to be determined [5]. In this context, an extensive immunogenomic analysis of more than 10,000 tumours comprising data from 33 diverse cancer types compiled by The Cancer Genome Atlas (TCGA) allowed to identify six different immune subtypes: wound healing, IFN-γ dominant, inflammatory, lymphocyte depleted, immunologically quiet and TGF-β dominant [6]. Their characterization was based on differences in macrophage or lymphocyte signatures, Th1:Th2 cell ratio, extent of intra-tumour heterogeneity, aneuploidy, extent of neoantigen load, overall cell proliferation, expression of immunomodulatory genes and prognosis. Notably, specific driver mutations correlated with lower (CTNNB1, NRAS, or IDH1) or higher (BRAF, TP53, or CASP8) leukocyte levels across all cancers. Future studies should investigate the link between specific genomic alterations and their contribution to the adaptation of the tumour microenvironment. As expected, in this classification GBMs were among the “lymphocyte depleted” subtype displaying a prominent macrophage signature, with Th1 suppressed and high M2 response [6].

Due to their large number in the tumour microenvironment, TAMs represent a key target for GBM immunotherapy and a range of immunomodulatory agents are currently being trialled in patients. For example, as TAMs critically depend on colony-stimulating factor-1 (CSF-1) for their survival, differentiation and proliferation, strategies to target TAMs in the clinic include CSF-1 receptor (CSF-1R) blockade [7,8]. However, despite having shown an effect on tumour growth in mouse models, this approach failed to improve overall survival in patients [9], suggesting that putative TAM subpopulations may be resistant to CSF-1R inhibition [10].

Despite extensive efforts in this direction, the precise role of TAMs in GBM onset and progression as well as how TAMs may affect current immunotherapeutic approaches, including vaccines, oncolytic viruses and immune-checkpoint inhibitors, remains unclear. Therefore, a deeper understanding of the complexity and diversity of TAM adaptive features is critical to develop novel personalized immune therapeutic strategies for GBM patients.

In this review, we will describe different features underlying TAMs heterogeneity and adaptation in GBM. Further, we will pinpoint the aspects linked to their diversity that warrant further investigations and how this heterogeneity may ultimately be harnessed for the development of novel personalized immune therapeutic strategies.

## 2. Tumour-Associated Microglia/Macrophages in Glioblastoma

### 2.1. M1 and M2 Polarization States: The Basic School of Thought

Macrophages are highly dynamic cells whose molecular profiles are substantially influenced by specific environmental cues. In vitro studies enabled classification of activated macrophages according to a binary system, with pro-inflammatory cytokines (e.g., IFNγ) skewing them towards a classical (M1-like) activation state, while anti-inflammatory cytokines (e.g., IL4) polarizing macrophages into an alternative (M2-like) phenotype [11]. A similar dual classification has been described in cultivated microglia exposed either to LPS/IFNγ or IL10/IL4 [12]. In cancer, this nomenclature has been used for decades to discriminate M1-like anti-tumour versus M2-like pro-tumour macrophages, with the latter assumed to constitute the majority of macrophages in the tumour according to their immune-suppressive properties [13]. However, this simplistic classification described in vitro does not apply to the in vivo situation as it only represents the two extremes of a continuum of activated states. Over time, further intermediate states describing M2-like macrophages have been introduced, with a putative M2c state associated to immune regulation, matrix deposition and tissue remodelling mostly observed in brain malignancies [14]. However, despite these efforts, little exclusivity was observed between these different categories of TAMs in GBM [15]. This is supported by studies conducted in GBM murine models where TAMs display an expression profile different from the predefined M1 and M2 polarization states, including a mixture of M1- and M2-specific genes [16] or analyses in GBM patients showing that TAMs exhibit a non-polarized M0 phenotype [17] (Figure 1). More recently, another attempt to distinguish between pro- and anti-tumour macrophages has been based on surface markers, where M1-like macrophages have been associated with the expression of CD40, CD74, MHC-II and phosphorylated STAT1, whereas M2-like cells express CD163, CD204, arginase-1 (ARG1) and phosphorylated STAT3 [14]. However, these markers have also failed to provide a robust separation and the subsequent understanding of their relative contribution to disease pathogenesis is still unclear.

### 2.2. Impact of Ontogeny on Tumour-Associated Microglia/Macrophage Functionality

The healthy brain harbours specific populations of tissue-resident macrophages effectively located in the parenchyma, perivascular spaces, meninges and choroid plexus where they maintain tissue homeostasis and ensure immune functions [18]. Within the parenchyma of the central nervous system (CNS), microglia are unique specialized immune effector cells that populate the brain early during embryogenesis [19]. In the adult brain, microglia continuously scan the environment and carry out several tasks, including neuronal support, phagocytosis of apoptotic cells and immune surveillance [20,21]. Their pool is maintained by self-renewal without contribution from bone marrow-derived progenitors, thus making microglia the only resident immune cell population in the healthy brain [22,23]. However, under certain pathological conditions, such as in GBM, the local inflammatory environment can compromise the integrity of the blood brain barrier leading to the infiltration of inflammatory monocytes from the circulation, which subsequently differentiate into monocyte-derived macrophages once they enter the brain tissue [24]. Therefore, in GBM, tumour-associated macrophages encompass resident parenchymal microglia, perivascular macrophages and peripheral monocyte-derived cells [25]. As a general observation, although tumour-associated macrophage proportions may vary in an organ-dependent manner, they have emerged as one of the most critical cell types contributing to worse prognosis across the vast majority of cancers [26].

In GBM, TAMs are recruited to the tumour site through various mediators, including CCL2, CX3CL1, CSF-1, GM-CSF and osteopontin released by neoplastic cells [24,27,28,29]. Upon accumulation at the tumour site, the functions of TAMs are supposed to be progressively overturned towards a pro-tumorigenic phenotype. For example, TAMs promote immune suppression and angiogenesis through the release of specific anti-inflammatory cytokines (e.g., TGFβ or IL10) and angiogenic factors (e.g., VEGFα) (see reviews [14,30]. Functionally, microglia and monocyte-derived macrophages react differently to various types of CNS insults [31] and the specific roles for these distinct cell populations are now starting to emerge in GBM. For example, it has been recently shown that the immune suppressive microenvironment in GBM patients depends on the accumulation of monocyte-derived macrophages [32].

Experimentally, approaches to distinguish resident microglia from other inflammatory immune cells entering the CNS have traditionally relied on CD45 expression to discriminate resident microglia (CD11b^+^CD45^low^ cells) from peripheral monocyte-derived macrophages (CD11b^+^CD45^high^ cells) [33]. However, this strategy has been recently challenged showing that glioma-associated microglia upregulate CD45 expression, thus limiting the effective discrimination of both populations in this disease [34]. Recently, using multiple genetic lineage tracing in transgenic (GEMM-shP53) and syngeneic GL261 mouse models, Bowman and collaborators have demonstrated that microglia specifically repress *Itga4* (CD49D), enabling the distinction between microglia and monocyte-derived macrophages in murine tumours [35] (Figure 1). Gene expression profiling demonstrated that both populations exhibit distinct activation states despite common traits of tumour education [35]. An unbiased meta-analysis of five published murine transcriptional datasets identified discriminatory marker sets distinguishing microglia versus peripheral monocytes/macrophages in health and gliomas [36]. These findings were validated at the protein level using syngeneic GL261 and RCAS-PDGFB driven GBM mouse models, where microglia-enriched genes included *P2ry12*, *Tmem119*, *Slc2a5* and *Fcrls*, whereas *Emilin2*, *Gda*, *Hp* and *Sell* were mainly expressed by peripheral monocytes/macrophages [36].

Further investigations will be critical to study how monocyte-derived macrophages in GBM influence the immunological functions of resident microglia. For example, during CNS injuries, peripheral macrophages affect nuclear factor kappa B (NFκB) signalling pathways in microglia reducing their phagocytic and inflammatory responses [37]. In cancer, targeting NFκB prompts TAMs towards a more cytotoxic anti-tumorigenic phenotype with a more activated state characterized by higher IL12 and MHC-II expression together with reduced levels of IL10 and ARG1 [38].

## 3. Tumour-Associated Microglia/Macrophages as Therapeutic Targets in Glioblastoma

### 3.1. Effect of Chemotherapy and Radiotherapy on Tumour-Associated Microglia/Macrophages

To date, the combination of radio-chemotherapy with immunotherapeutic agents has not been effective in GBM and drugs driving anti-tumour immune responses are currently evaluated in clinical trials. In principle, radiation can increase in situ immunogenicity of malignant cells, thus improving tumour immune recognition and T-cell mediated anti-tumour responses [39]. In these regimens, it remains to be determined what is the optimal radiation dose and schedule to harness the best immune effect. Moreover, it has to be considered that systemic administration of chemotherapeutic agents has immunosuppressive effects, thus representing a major challenge for effective anti-cancer immunotherapy-based strategies. In addition, high doses of glucocorticoids, such as dexamethasone, are usually administered to GBM patients to reduce inflammation and radiotherapy-induced cerebral oedema [40], thus dampening the inflammatory response by exerting profound effects on T cell subsets and NK cells [41]. Regarding TAMs, they are supposed to have a bimodal response to chemotherapy and radiotherapy, which can either reduce or amplify the magnitude of the anti-tumour responses [15]. These can be induced upon irradiation where targeted cancer cells generate damage-associated molecular patterns (DAMPs), such as high mobility group box 1 (HMGB1), that are recognized by pattern-recognition receptors (PRRs), including TLR2 and TLR4 in myeloid cells, that in turn trigger a pro-inflammatory phenotype [42]. Another route how radiation can induce anti-tumour immunity in immunogenic tumours is via STING and type I IFN-dependent signalling in dendritic cells [43]. It remains to be seen whether such mechanisms are active in immunologically ‘cold tumours’ such as GBM. Overall, it is evident that a thorough understanding of the complex interplay between tumour immunogenicity, the immune system and the adjuvant therapy will be critical to optimize and fine-tune the efficacy of immunotherapeutic approaches in GBM.

### 3.2. Depletion of Tumour-Associated Microglia/Macrophages in Glioblastoma

Upon accumulation to the tumour site, TAMs are thought to drive immune-suppression and promote tumour progression. Due to their high numbers in GBM, their genomic stability and adaptability to the microenvironment, several strategies to deplete TAMs have been developed. For example, liposome-encapsulated clodronate, which has been commonly used to deplete macrophage populations by inducing their apoptosis once phagocytosed by the cells, reduced tumour invasion in GL261 cultured brain slices, which was restored after addition of TAMs [44]. However, it has been recently demonstrated that intracerebral administration of clodronate liposomes into brain parenchyma can deplete microglia, but can also damage other brain cells and blood vessel integrity [45], therefore lacking specificity for TAMs. Further, attempts to specifically target peripheral macrophages, for example limiting monocyte infiltration via *Ccl2* genetic ablation, prolonged the survival of tumour-bearing mice [46], but these approaches have not been applied to patients yet. On the contrary, administration of ganciclovir to transgenic mice expressing thymidine kinase under the CD11b promoter reduced the CD11b^+^ population and contributed to 30% of tumour increase in the GBM syngeneic GL261 mouse model [47]. A major drawback of these studies is that these results were obtained in highly immunogenic GBM mouse models, while GBMs in patients are poorly immunogenic and display low T cell infiltration [48]. Further, TAMs depletion occurred prior to glioma cells implantation, therefore gliomagenesis may be substantially affected in the absence of TAMs. In silico studies have shown that TAM depletion therapy may be beneficial only for patients treated at early stages with a concomitant cytokine therapy [49].

These results highlight that depleting TAMs indiscriminately is probably not the optimal approach, as TAMs might play different roles depending on GBM features, including immunogenicity.

### 3.3. Immune Checkpoint Inhibitors and Reprogramming of Tumour-Associated Microglia/Macrophages in Glioblastoma

Immunotherapy is emerging as a promising approach holding great potential to foster tumour elimination by unleashing the immune system. The intense crosstalk between tumour cells, antigen presenting cells (APCs) and T cells is intricately controlled by multiple ligand-receptor interactions, known as checkpoints, which generally inhibit T-cell activation, ultimately affecting T cell cytotoxicity against tumour cells [50]. For example, the binding of PD-L1 expressed by tumour cells to its receptor PD-1 on T cells keeps the immune response in check. Hence, blocking this binding with an immune checkpoint inhibitor (e.g., anti-PD-L1 or anti-PD-1) enables T cells to attack the tumour cells. Similarly, the binding of APC-derived CD80/CD86 to CTLA-4 on T cells maintains the T cells in an inactive state and interfering with this binding allows T cells to be reactive. Evidently, the efficacy of T cell-based therapies is based on the amounts of tumour infiltrating lymphocytes (TILs), which are remarkably low in GBM [51]. Preclinical studies in the immunogenic GL261 syngeneic GBM mouse model have demonstrated the efficacy of targeting T cell immune-checkpoints, including CTLA-4, PD-1, PD-L1 and PD-L2 as monotherapies or in combination with radiotherapy [52]. However, as indicated above, this model poorly reflects human disease, since GBM patients typically show low mutational load and weak tumour immunogenicity, which correlates with poor response to immune checkpoint inhibitors [53]. The anti-PD-1 antibody advanced furthest in patients with GBM, however, in the phase III clinical trial, despite showing drug safety, it did not meet the primary endpoint of the study [5].

Due to TAMs abundance within GBM and their fast response to external stimuli, strategies to re-educate TAMs in mouse glioma models may be more efficient than their depletion or the use of immune checkpoint inhibitors. In this context, a promising target is signal-regulatory protein (SIRP) α, an inhibitory receptor expressed on myeloid cells that recognizes the CD47 ligand on tumour cells and contributes to immune evasion. The targeting of this axis with humanized anti-CD47 antibodies enhanced tumour phagocytosis and reduced tumour burden in patient-derived orthotopic xenografts of paediatric brain tumours [54]. Interesting results were also obtained using orthotopic xenografts and a syngeneic mouse model with genetically color-coded macrophages (*Ccr2*^RFP^) and microglia (*Cx3cr1*^GFP^), in which microglia were found to effectively phagocytose tumour cells in response to anti-CD47 blockade with a reduced inflammatory signature, making them a promising target for clinical applications [55]. Another example highlighting TAM subset-specific facets is the response to the VEGF neutralizing antibody bevacizumab, where blood-derived TAMs, instead of resident microglia, preferentially contributed to therapy resistance [56].

Combinatorial approaches targeting immune-suppressive populations concomitantly with promoting endogenous anti-tumour immune responses successfully impaired tumour progression in various subcutaneous tumour models. For example, dual targeting of suppressive myeloid populations by inhibiting CSF-1/CSF-1R signalling and activation of APCs with CD40 agonists conferred superior anti-tumour efficacy and increased survival compared with monotherapy. This effect was attributed to the decrease of immunosuppressive TAMs and Foxp3^+^ regulatory T cells as well as accumulation of tumour-infiltrating effector T cells exhibiting anti-tumorigenic features [57]. Further, the combination of an oncolytic virus expressing IL-12 together with the two immune checkpoint inhibitors anti-CTLA-4 and anti-PD1 was able to significantly reduce tumour growth in GBM intracranial mouse models [58]. Lastly, monotherapies or combinatorial approaches targeting TAMs are currently being undertaken in GBM clinical trials (Table 1).

Taken together, strategies aiming at reprogramming immunosuppressive myeloid cell populations and/or fostering anti-tumour immune responses in the tumour microenvironment may be necessary to empower checkpoint-based immune therapeutics in GBM. However, TAMs heterogeneity may represent a barrier to non-selective immunotherapies, which seek to target TAMs indiscriminately.

## 4. Dissecting Tumour-Associated Microglia/Macrophages Diversity at Single-Cell Resolution

### 4.1. Glioblastoma Subtyping and Single-Cell Analyses

In the last 10 years, multiple attempts have used transcriptional profiling to sub-classify GBMs into clinically meaningful tumour subtypes [59,60,61]. Although three common molecular subtypes (mesenchymal, classical and proneural) have been proposed in various studies, they poorly correlate with clinically relevant parameters, such as patient survival, except in a subgroup of patients [62]. Of note, the mesenchymal subtype was found to be characterized by a low tumour purity score along with an enrichment of TAMs, highlighting the contribution of the microenvironment in transcriptional profiling based on bulk tissue analysis. Furthermore, surgical multisampling has revealed that molecular subtypes can be present within the same patient tumour, suggesting that they do not represent bonafide subtypes, but rather reflect heterogeneous cellular expression programs [63]. More recently, this has been confirmed by single-cell RNA-sequencing (scRNA-seq) revealing the dynamic plasticity of GBM cells [64,65]. Hence, these studies highlight that tumour cells from a distinct GBM biopsy can display molecular traits reflecting different cellular states, a concept that is reminiscent of the M1 and M2 states in TAMs. Thus, at present the most promising classification strategies for gliomas are based on DNA methylation, allowing to discriminate IDH-mutant gliomas and IDH-wildtype gliomas [61,66].

### 4.2. Single-Cell Analyses of Microglia and Macrophages in Glioblastoma: Heterogeneity beyond Polarization States and Ontogeny?

Recent scRNA-seq studies also highlighted tissue-specific myeloid cell heterogeneity associated with distinct brain region-dependent transcriptional identities in health and disease [18,67,68]. For example, a specific disease-associated microglia subset localized around beta amyloid plaques has been described in Alzheimer’s disease [69]. The existence of distinct subpopulations of microglia, recently described under acute inflammatory conditions [70], suggests that different pools of microglia readjust their phenotype in response to environmental stimuli. Supporting this concept, studies conducted in neuroinflammatory diseases, including multiple sclerosis, have revealed the intricate heterogeneity of the myeloid compartment of the central nervous system along disease progression [71].

Likewise, the heterogeneity of TAMs in GBM is also starting to emerge. For example, scRNA-seq analyses of GBM biopsies demonstrated that TAMs frequently co-express canonical pro-inflammatory (M1) and alternatively activated (M2) genes in individual cells [72] (Figure 1). Further, in low grade gliomas, a gene signature of blood monocyte-derived TAMs, but not that of resident microglial TAMs, correlated with poor survival [72]. Similar studies provided insights about the spatial localization of TAMs. Correlation studies from a panel of established macrophage- and microglia-specific marker genes [73] enabled identification of a macrophage core signature highly present within the tumour core, while cells from the periphery expressed an evident microglia signature [74]. Additionally, pro-inflammatory markers (e.g., *IL1α* and *IL1β*) were highly expressed at the tumour periphery, while a more anti-inflammatory phenotype (e.g., *IL1RN*) was observed in the tumour core (Figure 2). Lastly, subpopulations within the tumour core seemed to promote vascular permeability and endothelial growth via the expression of *VEGFα* and an extracellular matrix remodelling gene signature [74].

The cellular composition of IDH-mutant gliomas was also unveiled by scRNA-seq, suggesting that astrocytomas (IDH-A) and oligodendrogliomas (IDH-O) share common lineages of glial differentiation with distinct tumour microenvironment signatures [75]. Specifically, a higher fraction of undifferentiated and cycling tumour cells was associated with enriched microglia/macrophage signatures in IDH-A, which correlated with tumour grade, thus providing a molecular fingerprint of tumour progression [76]. The authors propose that the composition of the tumour microenvironment may be driven by genetic influences, such as TP53, which is mutated in IDH-A, but not IDH-O gliomas, and TP53 has been shown to influence several immune pathways, including NF-kB [77].

In IDH-wildtype gliomas, it has been very recently shown that TAMs acquire a disease-associated signature related to aging microglia programs, including downregulation of the microglia homeostatic genes and upregulation of inflammatory, metabolic and interferon-associated genes. Various TAM clusters, including subsets enriched for positive regulation of vasculature development or antigen processing via MHC class I, have been identified [78] (Figure 1). Taken together, TAMs heterogeneity in glioma is currently emerging and should be taken into account when designing therapeutic approaches based on specific GBM features.

## 5. Conclusions and Perspectives

Microglia and macrophages in GBM are educated by the tumour and display unique molecular programs, which largely drive tumour-supportive phenotypes. However, the available subpopulations and functions of these cells along GBM development and progression are only partially understood. TAMs have been classified based on their activation state, their function and their morphology. With the advent of single cell analyses, it has become increasingly clear that the classification is complex and does not fully capture the heterogeneity of these cells in the context of GBM. Immunotherapeutic approaches in GBM will need to take into account the role of TAMs and their functional, spatial and temporal heterogeneity.

The local GBM microenvironment actively reprograms TAMs to establish new functional states with distinct gene expression profiles. If so, which TAM subsets arise during GBM development? Which subset of TAMs are more prone to be re-educated? Are their dynamic molecular states associated to TAM specific functions along GBM development and progression? Will their dissection help to improve TAM targeted therapies in combination with current treatment regimens? Taken together, it will be critical to address these questions to determine the most appropriate combinatorial approaches and to identify patient subgroups that may benefit most.

As perspectives, it will be fundamental to combine single-cell approaches, such as scRNA-seq and imaging mass cytometry, with functional screening of inferred cellular diversity, which will be critical to identify TAM subsets across GBM subtypes, landscapes and tumour stages, thus enabling targeting of putative pro-tumorigenic TAM subpopulations and/or to empower the anti-tumorigenic ones. Further, shedding light on the functional crosstalk between neoplastic cells and the tumour microenvironment at single-cell resolution will enable to dissect complex cell–cell interactions and how these may affect patient outcomes. Using a combination of scRNA-seq and flow cytometry in syngeneic mouse models of solid tumours allowed to profile potential cell-cell interactions between neoplastic and non-neoplastic cells [79]. Building precise cellular and molecular networks, which accurately reflect the complex and heterogeneous interactions between the tumour and immune elements, will open up avenues for novel combinatorial immunotherapies aiming at restoring an efficient immune response ultimately supporting the eradication of GBM.

## Figures and Tables

**Figure 1 ijms-21-00689-f001:**
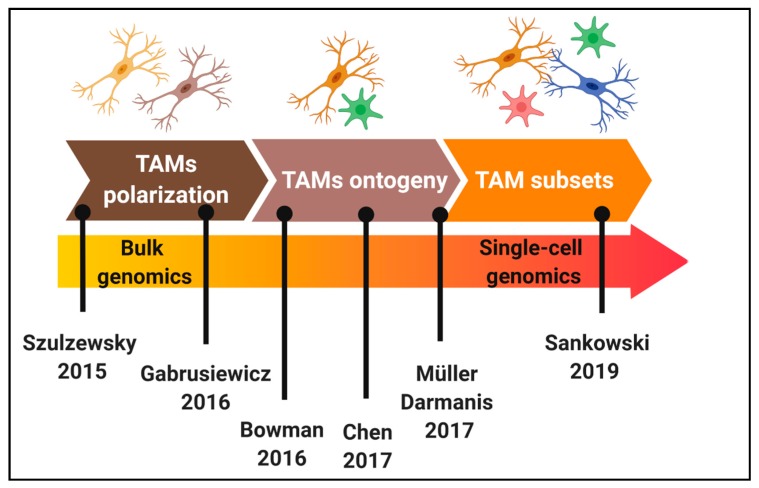
Chronology of the characterization of tumour-associated microglia/macrophages (TAMs) heterogeneity in glioblastoma (GBM). Key studies that have contributed to elucidate TAMs polarization, ontogeny and subsets in GBM mouse models and patients.

**Figure 2 ijms-21-00689-f002:**
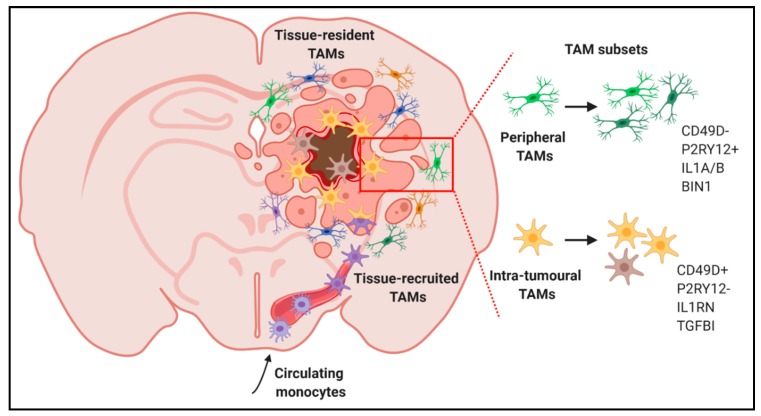
Discrimination of specific TAM subpopulations in GBM patients. TAMs present distinct features according to their ontogeny and spatial localization in specific tumour areas.

**Table 1 ijms-21-00689-t001:** Examples of current clinical trials targeting TAMs in GBM. RT: radiotherapy; TMZ: temozolomide; MRI: magnetic resonance imaging; LITT: laser interstitial thermal therapy; rGBM: relapsed/recurrent glioblastoma.

Myeloid Target	Drug Name	Additional Treatment	Study Phase	Tumour Type	Study Identifier
CSF-1R inhibitor	Cabiralizumab	Nivolumab (anti-PD-1)	I	GBM	NCT02526017
CSF-1R inhibitor	BLZ945	PDR001 (anti-PD-1)	I/II	GBM/rGBM	NCT02829723
CSF-1R inhibitor	Pexidartinib	RT + TMZ	I/II	GBM	NCT01790503
CXCR4 inhibitor	USL311	Lomustine	II	rGBM	NCT02765165
PD-L1 inhibitor	Avelumab	MRI-guided LITT therapy	I	rGBM	NCT03341806
STAT3 inhibitor	WP1066	-	I	rGBM	NCT01904123
GM-CSF	VBI-1901	-	I/II	rGBM	NCT03382977
MIF inhibitor	Ibudilast	TMZ	I/II	GBM/rGBM	NCT03782415

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
