# Peer review of "Revealing and Harnessing Tumour-Associated Microglia/Macrophage Heterogeneity in Glioblastoma"

_ijms, 2020, doi:10.3390/ijms21030689_

Round 1

Reviewer 1 Report

The authors provided a concise review on recent progress on tumor-associated microglia/macrophages in glioblastoma together with well-selected citations. However, the authors should modify the manuscript to increase the value of the paper.

#1. The authors discussed the phenotypical and functional differences between tumor-associated microglia and macrophages in more detail.

#2. It is better to change the paragraph from lines 158 to 185, because most of the paragraph is not directly related to glioma treatment. Thus, the authors should focus the discussion on the studies which are directly related to glioblastoma.

#3. If presently available, information on the present status of clinical trials targeting TAMs can enhance the value of the review.

Author Response

Reviewer #1

The authors provided a concise review on recent progress on tumor-associated microglia/macrophages in glioblastoma together with well-selected citations. However, the authors should modify the manuscript to increase the value of the paper.

We thank the reviewer for the positive and constructive evaluation of our manuscript.

#1. The authors discussed the phenotypical and functional differences between tumor-associated microglia and macrophages in more detail.

We thank the reviewer for this suggestion. The information on this subject in Glioblastoma is to some extent still scarce. We added one recent key article describing that the immune suppressive microenvironment in Glioblastoma patients depends on the accumulation of bone marrow-derived macrophages (line 131).  

#2. It is better to change the paragraph from lines 158 to 185, because most of the paragraph is not directly related to glioma treatment. Thus, the authors should focus the discussion on the studies which are directly related to glioblastoma.

We agree with the reviewer’s comment highlighting that most of the experimental citations in this paragraph are not directly related to Glioblastoma. For this, we removed the description of the studies conducted in pancreatic carcinoma, breast and lung cancers. However, although providing a short glimpse to this subject, we believe that this paragraph (3.1) underlining the importance for immunotherapies approaches to take into account the effect of the actual therapy regimen on TAMs should be maintained.

#3. If presently available, information on the present status of clinical trials targeting TAMs can enhance the value of the review.

We thank the reviewer for this recommendation. We added this important information in Table 1 at the end of paragraph 3.3 as several clinical trials in GBM deal with the combination of immune checkpoint inhibitors and myeloid targets.

Reviewer 2 Report

The manuscript is well-written, structured, analytical and balanced. However, I would recommend (if the additional data exist) to more detail on functional heterogeneity of microglia/macrophage in GBM.

Line 102: “is still uncle” should be corrected (typo).

Author Response

Reviewer #2

The manuscript is well-written, structured, analytical and balanced. However, I would recommend (if the additional data exist) to more detail on functional heterogeneity of microglia/macrophage in GBM.

We thank the reviewer for the positive feedback of our manuscript. As for Reviewer #1, the information on functional heterogeneity of microglia/macrophage in GBM is to some extent still scarce. We added one recent key article describing that bone marrow-derived macrophages contribute to the immune suppressive microenvironment observed in Glioblastoma patients (line 131).  

Line 102: “is still uncle” should be corrected (typo).

The typo has been corrected.